# Hot Spots in Urogenital Basic Cancer Research and Clinics

**DOI:** 10.3390/cancers17071173

**Published:** 2025-03-31

**Authors:** Claudia Manini, Gorka Larrinaga, Javier C. Angulo, José I. López

**Affiliations:** 1Department of Pathology, San Giovanni Bosco Hospital, ASL Città di Torino, 10154 Turin, Italy; claudia.manini@aslcittaditorino.it; 2Department of Nursing, Faculty of Medicine and Nursing, University of the Basque Country (UPV/EHU), 48940 Leioa, Spain; gorka.larrinaga@ehu.eus; 3Department of Physiology, Faculty of Medicine and Nursing, University of the Basque Country (UPV/EHU), 48940 Leioa, Spain; 4Biobizkaia Health Research Institute, 48903 Barakaldo, Spain; 5Clinical Department, Faculty of Medical Sciences, European University of Madrid, 28905 Getafe, Spain; javier.angulo@salud.madrid.org

**Keywords:** urogenital cancer, kidney cancer, urinary tract cancer, prostate cancer, testicular cancer, penile cancer, histology, molecular analysis, diagnosis, treatment

## Abstract

Urogenital cancer is very common in the male populations of Western countries, a problem of major concern for public health systems, and a frequent test bench for oncological research. In this narrative, we identify the main hot topics on clinics and the basic science of urogenital cancer in the last few years (from 2021 onwards), considering the information given in the abstracts of almost 300 original articles published in outstanding journals of pathology, urology, and basic science. Once defined, each issue in the top ten list of hot topics is independently reviewed in order to put together the most relevant updates and/or useful features, accompanied by selected illustrations. The main recipients of this article are clinicians (pathologists, urologists, and oncologists).

## 1. Introduction

Urogenital cancer leads the ranking of the most common neoplasms in Western countries and represents one of the most active test areas for basic and clinical cancer research today. The estimation of new cases in the USA in 2024 includes prostate, urinary bladder, and kidney cancer in the top ten list of the most frequent tumors, amounting to 40% of malignancies in men, and the expected death rate from prostate and bladder cancer in the male US population in the same period is up to 15% [1]. Prostate cancer (PCa) leads the list of new cases and is the second-most common tumor of estimated deaths. GLOBOCAN estimates show similar figures in Europe [2].

Renal cell carcinoma (RCC), with almost 200,000 deaths per year, is the eighth most common cancer in the US. Tobacco, obesity, and hypertension are well-recognized risk factors. Urinary bladder carcinoma (BC) is the most common tumor in the urinary tract and represents the eleventh most frequent malignancy in the general population (males and females). Occupational exposure to diverse compounds, particularly chlorate hydrocarbons, aromatic amines, and polycyclic hydrocarbons, is a well-known risk factor. PCa is the most common malignancy in men, with almost 300,000 new cases, and is the second cause of deaths by cancer, with more than 35,000 deaths in the USA. Aside from genetic predisposition and race, there have been no evident risk factors detected so far. Penile cancer (PeCa) is quite common in less developed regions of the world, but it is a rare neoplasm in the USA, accounting for less than 1% of cancers in men. Well-known risk factors are human papillomavirus (HPV) infection, tobacco, phimosis, and immunosuppression, i.e., HIV infection. Although testicular cancer (TC) also shows figures under 1%, it is the most common malignancy in young men.

Table 1 identifies the list of trending hot topics in urogenital oncology found in the literature review.

## 2. 2022 WHO Update on the Pathologic Classification of Urinary and Male Genital Tumors

### 2.1. Prostate Cancer

The 2022 WHO classification of PCa [3] recognizes PIN-like prostate adenocarcinoma as a new form of low-grade acinar adenocarcinoma (Gleason score = 6). Cribriform and/or papillary structures are not a feature in this tumor subtype, which typically presents mutations in *RAF*/*RAS*. PIN-like adenocarcinoma must be distinguished from ductal adenocarcinoma. The term ductal adenocarcinoma is maintained only in radical prostatectomy specimens with cancers containing more than 50% of ductal morphology. Then, the term “adenocarcinoma with ductal features” is recommended for cancers detected in core biopsy specimens. Ductal adenocarcinoma may show a wide spectrum of mutations (*ERG*, *SPOP*, *FOXA1*, *CTNBB1*, *APC*, etc.).

The 2022 WHO update [3] recognizes the neuroendocrine carcinoma secondary to antiandrogenic treatment. To note, this tumor variant develops in up to 17% of the patients treated with androgenic blockade (abiraterone, enzalutamide, etc.). Synaptophysin and/or chromogranin immunohistochemistry are not recommended for its identification in clinical practice, and most are positive with TTF-1, p53, PSA, and prostatic acid phosphatase. Notably, PSA and prostatic acid phosphatase are usually negative in treatment-related neuroendocrine carcinoma.

The presence of a cribriform pattern worsens the prognosis of Gleason scores 3 + 4, 4 + 3, and 4 + 4. Also, large (>12 lumina) cribriform structures (Figure 1) negatively impact the prognosis of PCa.

Finally, the term “basal cell carcinoma” is no longer used and is changed to “adenoid-cystic carcinoma” based on specific molecular findings.

### 2.2. Renal Cancer

New renal tumor entities have been included in the long list of RCC [4], i.e., eosinophilic solid and cystic RCC (ESC RCC) (Figure 2), *ELOC*-mutated RCC, *ALK*-rearranged RCC, *SMARCB1*-deficient medullary RCC, *TFEB*-altered RCC, and *FH*-deficient RCC.

Papillary RCC (PRCC) classification has been fully rearranged. The formerly called PRCC type 1 is not a unique entity and includes several new variants, i.e., renal neoplasm with inverted polarity (mutated in *KRAS*), psammomatous hyalinizing RCC (mutated in *NF2*), biphasic alveolo-squamoid renal RCC, and thyroid-like follicular RCC (*EWSR1*-*PATZ1* fusion). On the other hand, the “old” PRCC type 2 is no longer a specific entity and has been broken down into several new ones, i.e., *FH*-deficient RCC, tubulo-cystic RCC, eosinophilic solid and cystic RCC (ESC RCC), *SMARCB1*-deficient medullary RCC, and RCC of the MiTF group.

Several neoplasms have emerged within the spectrum between renal oncocytoma and chromophobe RCC, i.e., *SDH*-deficient RCC, eosinophilic vacuolated tumor, and low-grade oncocytic tumor.

Finally, the molecular classification of RCC in the 2022 WHO classification of renal tumors is still under construction.

### 2.3. Urinary Tract Cancer

The 2022 WHO classification of urinary tract carcinoma [3] maintains the division between high and low grades, thus correlating with the cytologic approach of the Paris system [5].

It is not mandatory (although advisable when possible) to define the level of invasion (pTa vs. pT1) in the low-grade non-invasive papillary urothelial carcinoma (UC). It is important to try to distinguish between inverted growth and true invasion.

The term “papillary urothelial neoplasia of low malignant potential” is maintained, and the term urothelial dysplasia is retained, but dysplasia is not a synonym of in situ carcinoma.

The following histological subtypes of UC are specifically recognized:-Tubular UC-Large “nested” UC (Figure 3)

Conceptual or terminological modifications in pre-existing tumor subtypes are the following:-Clear cell UC is renamed as “glycogen-rich” clear cell UC (Figure 3) to distinguish it from the adenocarcinoma of Müllerian type.-Plasmacytoid UC should no longer be termed as “signet-ring cell”/“diffuse”.-Glandular, squamous, trophoblastic (Figure 3), Müllerian, and neuroendocrine morphologies must be specifically mentioned in the pathological report of UC, including its approximate percentage.-Micropapillary, plasmacytoid (Figure 3), and other “wolves in lamb clothes” must be recognized and specifically reported, either pure or mixed, in the pathological report of UC.

Still non-solved issues in the UC diagnosis are:-The sub-staging of pT1 carcinomas (pT1a/pT1b).-The applicability of the molecular classification (luminal, basal, basal-squamous, etc.) in muscle-invasive UC.-The clinical usefulness of *FGFR*3, *TP*53, and *ERCC*2 mutations.

### 2.4. Testicular Cancer

The testis is a territory conditioned by extreme histological variability. The 2022 WHO classification of TC [4] recommends not to speak of “variants” but of types and subtypes because “variant” usually refers to genetic mutations. In general terms, the subclassification is maintained with respect to previous editions, although sex-cord tumors and stromal tumors, non-gonadoblastoma, are no longer accepted in this edition.

Seminoma is included in the group of germinomas. Indeed, seminoma with syncytiotrophoblastic cells deserves to be individualized as a specific subtype, while the rest of the seminomas do not. The group of non-seminomatous germ cell tumors does not change (yolk-sac tumor, embryonal carcinoma, and trophoblastic tumors).

Consider using mm^2^ as the surface unit for mitotic counting, and not high-power fields, to avoid possible variations in the magnification from one microscope to another.

The term PNET must not be used in the group of teratomas with somatic transformation, neither in the testis nor in the ovary, to avoid any misunderstanding with the PNETs in the central nervous system; instead, the name ENET (embryonal neuroectodermal tumor) has been coined for these cases.

Within the group of germ cell tumors not related to in situ germ cell neoplasia, the term spermatocytic tumor (Figure 4) is maintained. To note, the name “carcinoid tumor” is no longer used. A testicular neuroendocrine tumor is renamed as a “prepuberal testicular neuroendocrine tumor”, since most of them arise in prepuberal teratomas. The sertoliform cystadenoma moves from the adnexal tumors group to the cluster of Sertoli cell tumors.

Newly recognized entities in this edition are:-Stromal tumor with signet-ring cells.-“Myoid” gonadal stromal tumor.-Well-differentiated papillary mesothelial tumor (Figure 4).

### 2.5. Penile Cancer

The 2022 WHO classification of PeCa [4] distinguishes HPV-related and HPV-unrelated carcinomas.

-HPV-related squamous cell carcinomas are:
▪“Warty” PeCa (Figure 5).▪Basaloid PeCa.▪Clear cell squamous PeCa.▪Lymphoepithelioma-like PeCa.
-HPV-unrelated squamous cell PeCa are:
▪Conventional squamous PeCa, either pseudo-hyperplastic or acantholytic/pseudo-glandular.▪Verrucous PeCa, including carcinoma “cuniculatum” (Figure 5).▪Papillary squamous PeCa (Figure 5).▪Sarcomatoid squamous PeCa.
-Squamous cell PeCa, not otherwise specified.-Mixed squamous cell PeCa.

## 3. New Entities in Kidney Cancer

RCC has become, in the last decade, a neoplasm with an increasingly complex constellation of clinical, histological, and molecular features, representing one of the hottest and most exciting topics in pathology. In this particular context, the Genitourinary Pathology Society (GUPS) has brought order to this puzzle, updating the last findings in the existing WHO RCC in 2021 [6], as well as an update of the list of renal tumor novelties [7]. New tumors include novel eosinophilic solid and cystic RCC, ALK rearrangement-associated RCC, and RCC with fibromyomatous stroma (Figure 6), emerging (eosinophilic vacuolated tumor and thyroid-like follicular RCC), and provisional (low-grade oncocytic tumor, atrophic kidney-like lesion, and biphasic hyalinizing psammomatous RCC) tumor entities. A detailed compiled description with a literature review of all these tumors is provided in the aforementioned GUPS article [7].

## 4. Urinary Cancer-Omics

Thousands of studies have been implemented in the last decade, analyzing genomics, proteomics, transcriptomics, and other -*omics* in urological cancer, providing interesting and promising data for patient management. Here, we overview some hot spots in renal, urinary tract, prostate, and penile cancers.

### 4.1. Renal Cancer

Aside from the well-known *VHL* inactivation responsible for clear cell renal cell carcinoma (CCRCC) genesis, deletions of chromosome 3p also involve other suppressor genes like *PBRM1*, *SETD2*, and *BAP-1*. Interestingly, *PBRM1* and *BAP-1* mutations in CCRCC are associated with specific clinical courses, attenuated and aggressive, respectively [8]. Other implicated genes include histone-modifying genes like *KDM5C* and *KDM6A*, and mTOR pathway genes like *TSC1*, *TSC2*, *MTOR*, *PIK3CA*, *PTEN*, and *TP53*. One of the most important limitations in defining the molecular profile of CCRCC is its inherent high levels of intratumor heterogeneity (ITH), by which different regions of the same tumor display different molecular alterations [9] associated with specific prognosis and clinical aggressiveness. This characteristic has made CCRCC a perfect test bench for analyzing tumor complexity [10].

The genomics of PRCC include *MET* oncogene upregulation in up to 80% of the cases, a gene that has raised different clinical attempts for targeted therapies. Mutations on *TERT*, a gene malfunction also involved in many other cancers, including UC, have also been detected in PRCC, where it has been associated with high metastatic capability and bad prognosis. Other less frequently mutated genes in PRCC are *CDKN2A* and *CDKN2B*.

Chromophobe renal cell carcinoma (ChRCC) displays a lower mutation frequency compared with the precedent RCC, with *TP53*, *TERT*, *mTOR*, and *PTEN* genes being the most frequently involved.

*NF2*, *FBXW7* tumor suppressor, and *CDKN2A* are genes whose malfunctioning has been detected in collecting-duct RCC, a rare and aggressive form of RCC.

Aside from that, an important practical issue is to know the molecular alterations with clinical significance in RCC, a promising perspective in terms of conducting future molecular classification of renal neoplasia [6,7]. Among them, *TFE3*-rearranged RCC, t(6;11) translocated (*TFEB*) RCC (Figure 7), *FH*-deficient RCC, *SDH*-deficient RCC (Figure 8), *SMARCB1*-deficient medullary RCC, ALK-rearranged RCC, and ELOC-mutated RCC are included.

### 4.2. Prostate Cancer

Localized PCa shows multiple genomic and proteomic alterations. A single-cell proteomic study revealed that some cases, at the early stage, already contained subclones typically detected in castration-resistant and metastatic cases [11].

Focusing on the advanced PCa, many studies have confirmed the marked multifocality and ITH, both histologically (Figure 9) and genetically, of the primary PCa itself. This condition has been evidenced by analyzing multi-region sampling in radical prostatectomy specimens and in their paired metastases, and in this way identifying significant differences in the gene expression between different tumor areas. For example, whole-exome sequencing analyses of primary and metastatic PCa samples have demonstrated that androgen receptor (*AR*) gene-activating alterations were the most frequent genomic event found in the metastases, followed by *TP53* mutations and *MYC* amplifications. Interestingly, *AR* alterations were not present in their paired primary tumors. The *AR* activations found in the metastases of many PCa may be a sign of an over-imposed castration resistance status. *FOXA1*, *KMT2C*, and *KMT2D* are other genes the alterations of which are more frequent in the metastases than in the primary tumors. *BRCA2* germline alterations have been detected in a minority of cases. Since *PTEN* loss has been previously associated with progression and metastases, some authors have proposed considering the immunohistochemical PTEN loss of expression as a reliable marker of clinical aggressiveness in PCa [12].

### 4.3. Urinary Tract Cancer

The molecular subtyping of UC [13] distinguishes several tumor types with different clinical significances, with the luminal (including luminal papillary and luminal infiltrating) and basal (including squamous) subtypes being the most common ones (Table 2).

Papillary UC—derived from intermediate cells in the urothelium—and UC in situ—derived from basal cells in the urothelium—follow two different pathways on urothelial carcinogenesis with specific clinicopathological and molecular features.

Proteogenomic analyses have clarified the distinction between urothelial papilloma and papillary UC (Figure 10), a classic differential diagnosis under a light microscope. Therefore, urothelial papilloma shows a higher mutation rate in *MPRIP*, *HRAS*, and *MAP3K1* genes, whereas papillary UC carries mutations in *FGFR3* and *PPFIBP1*. Of them, *HRAS* mutations have been reported to be the predominant inverted type of urothelial papilloma. HRAS, ALDH7A1, CBLB, and MPRIP proteins are differently expressed in urothelial papilloma and papillary UC.

DNA damage related to the APOBEC signature is a key pathway in in situ progression to invasive forms of UC, a fact that occurs in roughly 60% of cases. Approximately 15% of UC patients develop distant metastases, a fact probably associated with *RBPMS* losses through the activation of AP-1 transcription factors.

### 4.4. Penile Carcinoma

Squamous PeCa displays a wide spectrum of mutations [14], with *TP53*, *TERT* promoter, and *PIK3CA* among the most frequently identified. Furthermore, *CDKN2A* losses and CCND1 amplifications have also been detected in a small number of cases.

## 5. Update on Gleason Grading System

The Gleason system for PCa grading was proposed by Donald Gleason in 1966. It has demonstrated high levels of inter- and intra-observer reproducibility for decades among pathologists and an optimal correlation with the prognosis of PCa patients. It has been updated several times in light of new histological findings and clinical evidence from large-cohort studies. Changes included in the last 2022 WHO update [3] have been previously mentioned in this article.

Prognostic groups in PCa have been proposed in the last decade [15]. The goal was to eliminate some misunderstanding; for example, if the lowest possible Gleason score in clinical practice is 3 + 3 = 6 (Figure 11) and the highest is 5 + 5 = 10 in a range from 2 to 10, which means that a sum of 6 might represent an erroneous perception of an intermediate grade, with unnecessary stress for patients. Another imprecision concerns a Gleason score of 7, because clinical evidence has shown that 3 + 4 is not the same as 4 + 3. For interested readers, precise definitions and additional reasons for the generalized adoption of these prognostic groups have been recently reviewed elsewhere [16,17].

In recent years, the question has arisen as to whether prognostic group 1 PCa (Gleason score 3 + 3) (Figure 11) should be considered cancer or not [18]. From a clinical viewpoint, there are some reasons for reconsidering this diagnostic category; for example, the well-known indolence of these cases, the risk of overtreatment, the possibility to adopt an active surveillance on these patients, and the unnecessary stress generated on patients when hearing the word “cancer”. However, the general consensus of pathologists is to continue considering these cases as low-grade cancers in core biopsies [19] for several reasons. First, core biopsy may not reflect the reality of the tumor because high-grade prostate cancer may not be sampled even when using appropriate protocols. Second, although infrequent, the possibility of tumor progression in the form of extraprostatic extension and metastatic spread does persist in these cases. Third, Gleason score 3 + 3 = 6 prostatic adenocarcinoma maintains the immunohistochemical staining and molecular changes of high-grade PCa.

## 6. Targeted Therapies and Other Novel Treatments in Urologic Cancer

This is an extremely dynamic area in terms of updates through numerous trials. The classical pillars of cancer care include surgery, radiotherapy, cytotoxic chemotherapy, and precision (molecular targeted) therapy. However, immunotherapy has emerged as a new (re)volution in the standard of care in certain patients with genitourinary malignancies. While the benefit of classical immunotherapy obtained in the past for metastatic CCRCC has been based on interferon-α and IL-2, and early-stage bladder UC using Bacillus Calmette-Guerin therapy has been modest, therapies with immune checkpoint inhibitors demonstrate meaningful survival benefit and durable clinical response in RCC, UC, and some patients with PCa [20]. The combination or sequencing of immunomodulators, either with chemotherapy or other targeted therapies, is the current hot topic in urologic cancer therapy, not only for metastatic cancer but also in the early disease, either as a neoadjuvant or adjuvant setting. Interested readers are advised to address the contemporary information in every tumor type. Here, only general outlines for non-specialists are mentioned.

Partial or radical surgery remains the first elective treatment in RCC. In the metastatic stage, there is a great option of therapies, including tyrosine kinase inhibitors, such as sunitinib, axitinib, or cabozantinib, to block VEGF pathways and mTOR receptors, alone or in combination with immune checkpoint inhibitors such as nivolumab, ipilimumab, or pembrolizumab [21]. Although this wealth of options has established RCC at the forefront of solid tumor immunology, many questions still remain unanswered [22]. An open-label, randomized phase 3 study in 2022 revealed a higher overall survival of nivolumab plus ipilimumab compared with sunitinib alone in advanced RCC [23], but the issue of which is the best therapy is still unsolved, as synergistic toxicity needs to be minimized. Furthermore, many patients treated with immune checkpoint inhibitors will experience disease progression, and a standardized second-line approach in these cases remains unknown, as a rechallenge with combined immunotherapy after early progression is not recommended [24]. Although precision immunotherapy is a tremendous promise in advanced RCC, there has unfortunately been limited success thus far. A deeper understanding of the tumor immune microenvironment in RCC could help to target newer immune checkpoints to generate a more vigorous anti-tumor response in the near future. Furthermore, the incorporation of mass spectrometry and single-cell sequencing technologies to develop immunogenomic tools will enable the discovery of newer tumor antigens serving as targets for chimeric antigen receptor (CAR) T cell therapies, the same as in hematologic malignancies [25]. Several other strategies have the potential to improve immunotherapy and could be used in the not-so-distant future. They include targeted lactate-lactylation and the modulation of the microbiome to potentially improve responses to immunotherapy [26,27,28].

Surgery plus Cisplatin-based chemotherapy remains the elective treatment in muscle-invasive bladder UC, and intravesical Bacillus Calmette-Guerin (BCG) instillations are still considered the best option in non-muscle-invasive (pTa/pT1) high-grade bladder UC [29]. However, BCG fails in nearly 40% of patients, thus requiring alternative treatments. Traditionally, radical cystectomy (which severely impacts the quality of life) has been the standard treatment for BCG-unresponsive disease, although recent advances have focused on bladder-preserving therapies that leverage immune checkpoint inhibitors, viral gene therapies, novel drug delivery systems, and other targeted molecular agents. Very recent immune checkpoint inhibitors, such as pembrolizumab and durvalumab, and other immunomodulators have demonstrated potential for systemic treatments in BCG-unresponsive non-muscle invasive disease [30,31,32]. Furthermore, the role of platinum-based chemotherapy in the evolving treatment landscape of advanced UC must be reviewed. Immune checkpoint inhibitors and antibody-drug conjugates have proven survival benefits in metastatic contexts and are expanding their therapeutic applications to the perioperative setting for non-metastatic muscle-invasive disease, either as substitutes or in combination with cytotoxic chemotherapy [33,34]. Other therapeutic models like the oncolytic adenovirus XVir-N-31 are being investigated in bladder UC [35].

The therapeutic approach to PCa is complex, spanning from active surveillance to radical surgery or radiotherapy and hormone blockade. Several factors influence therapeutic decisions, such as age and patient comorbidities, stage, Gleason index, and histological risk group, among others. Immunotherapy is not an option thus far, and newly targeted therapies are emerging and require further investigations. The heterogeneity of the genomic landscape of metastatic castration-resistant tumors makes the advances in this issue very difficult. This heterogeneity and the complexity of the tumor immune microenvironment in late forms of advanced and metastatic PCa have brought a delay in establishing immunotherapy as a standard option in this serious disease. Ipilimumab and olaparib have proved to prolong survival compared to placebo but are still far from being incorporated into clinical practice. More recent options such as PSMA-targeted treatments and other PARP inhibitors are currently being evaluated [36]. Integrated machine learning could facilitate extensive analysis to identify epithelial cell marker genes that could be used to enhance immunotherapy in PCa [37].

Targeted therapies are still under investigation in penile cancer. Specifically, the analysis of PD-L1 expression seems to be higher in the pT2 stage and correlates with regional lymph node metastasis [38]. Currently, several trials are analyzing the potential use of avelumab [39] and retifanlimab [40] in this disease.

## 7. News on Non-Muscle-Invasive Urothelial Carcinoma

Non-muscle-invasive neoplasm is the most frequent form of UC in Western countries [1]. Main issues of discussion in the literature are histological grading and the precise histological definition of basal membrane disruption, that is, the distinction between pTa/pTis (intraepithelial) and pT1 (lamina propria/submucosa invasion), and the clinical approach to the disease; in other words, how to manage these patients using the armamentarium currently available.

From a pathology viewpoint, the identification of stromal microscopic invasion (disruption of the basal membrane) is still a sliding issue in non-muscle-invasive UC, as well as in squamous cell carcinomas of other topographies, because many of these tumors grow into the lamina propria with a “pushing” border where no isolated cells and/or small tumor cell infiltrating nests are identifiable. This problem is old [41,42] and still persists today. For a tumor, once defined as superficially (non-muscle) invasive, its level of invasion, either into the lamina propria or submucosa, matters in terms of prognosis [43,44]. Several methods based on microscopic analysis have been designed, but the misorientation of tissue fragments and the inherent artifacts related to transurethral resection procedures make such methods poorly reproducible. Although this issue has been recently revisited in a single-institution study [45], it remains unresolved so far.

UC grading has received a lot of attention since 1973, when a unified WHO grading system was adopted [46]. This WHO system has been updated several times, and comparisons between different versions of this system have been published [47,48].

Molecular analyses of UC have recently received full attention, showing significant different routes between in situ UC and papillary intraepithelial UC [13]. More specifically, *CCDC 138* is the most frequently mutated gene in situ UC (pTis) [49]. The topic is too broad to be condensed here, so the interested reader is invited to address the specific literature.

The therapeutic spectrum of non-muscle-invasive UC has been very recently reviewed, considering low-risk, intermediate, and high-risk, BCG-naïve, and BCG-unresponsive non-muscle-invasive variants [50].

## 8. Artificial Intelligence in Urologic Cancer

Diverse tools of artificial intelligence have been applied in urological cancer studies to help diagnosis, especially in PCa [51,52,53], to predict treatment response, to select patients for specific therapies, and, finally, to predict clinical evolution [54].

The identification of targetable molecular alterations [55] and histological data with prognostic implications like renal capsule invasion in RCC [56] have been unveiled by using radiomics models. Indeed, several recent studies have shown that machine learning models seem to be useful in predicting the risk of recurrence in non-metastatic localized RCC, a fact that might help in the selection of patients for personalized therapies [57,58].

These tools have also been applied, for example, to predict lymph node metastases [59] or HER2 status [60] in the UC of the urinary bladder, and to predict the histological composition of testicular tumors [61] and even to discriminate between the benign or malignant nature of testicular masses [62].

Artificial intelligence tools have also been implemented in PCa, mainly for the diagnostic purposes of core biopsies using morphological and immunohistochemical parameters [51,52,53]. An innovative approach combining diagnostic immunohistochemical markers (cytokeratins, p63, and racemase) and a customized segmentation network [63] has been developed to discriminate between PCa, prostate high-grade intraepithelial neoplasia (HG-PIN), and benign tissue in prostate core biopsies. This methodology would be potentially helpful to assist pathologists in the routine diagnosis of a quickly growing diagnostic area that highly impacts pathology lab burdens.

## 9. Intratumor Heterogeneity Influences Therapeutic Failures in Urologic Neoplasms

Massive sequencing studies [64,65] have demonstrated that ITH is an inherent condition in the evolution of cancer. Multi-region studies [8,66] have discovered the intricacies and particularities of every tumor type, so we already know that temporo-spatial ITH may develop in a diverse manner depending on the tumor and on the patient; in brief, ITH is a tumor- and patient-specific event that makes every neoplasm a unique and unrepeatable process. This acknowledgment is at the basis of the so-called personalized therapy and, at the same time, justifies therapeutic failures if the tumor is not deeply analyzed and ITH is correctly unveiled.

CCRCC is a good example in terms of analyzing ITH (Figure 12) from histological, immunohistochemical, and molecular points of view. Interestingly, specific clonal and subclonal mutations are linked with different tumor evolutions, metastatic potential, clinical course, and prognosis [8,67].

Multi-regional studies have also detected immunohistochemical ITH in muscle-invasive [68] and non-muscle-invasive [69] UC, as well as in PCa [70] and in non-seminomatous germ cell tumors of the testis [71].

Recent studies have demonstrated that resistances to therapy are closely linked to the presence of ITH [72], which, in this context, can be considered a survival response of tumor cells following ecological principles. The ecological perspective of ITH involves not only tumor cells but also the tumor microenvironment [73]. This fact makes the understanding of tumor biology and the implementation of successful targeted treatments even more complex.

Advances in single-cell RNA sequencing (scRNA-seq) have been crucial in unveiling the heterogeneity of both tumor cells and tumor microenvironments [74]. Complementary techniques like spatial transcriptomics [75], multiplex immunohistochemistry [76], and lipid imaging mass spectrometry [77] provide critical insights into the localization, density, spatial interactions, and metabolic-functional states of the diverse cells involved within the tumor ecosystem. Furthermore, computational tools are being developed to integrate these datasets [78]; for instance, Tumoroscope, a recently described method tested in PCa, enables comprehensive analysis of the genetic, functional, and spatial architecture of tumors, facilitating a deeper understanding of ITH and tumor evolution [79].

Understanding the interactions between tumor cells and the tumor microenvironment provides valuable opportunities to refine the classification of urological tumors, evaluate progression risk, and guide more precise therapeutic decisions [74]. In the era of immunotherapy, one of the most intensively studied biological phenomena is the generation of immunosuppressive microenvironments as a consequence of these interactions [80]. Immunosuppressive players such as M2 macrophages, myeloid-derived suppressor cells (MDSCs), and regulatory T cells (Tregs) orchestrate these environments, which also harbor immunologically exhausted cytotoxic T lymphocytes [81]. In turn, cancer-associated fibroblasts (CAFs) can hamper the infiltration of cytotoxic cells into the tumor through the extracellular matrix remodeling, induce M2 macrophage polarization, and drive the differentiation of naïve T cells into Tregs, collectively fostering a microenvironment conducive to tumor immune evasion [82].

These microenvironmental interactions often occur in different ways in distinct tumor regions [74]. In CCRCC, a paradigmatic example of ITH [67], synergies between specific Treg populations and M2 macrophage polarization have been observed predominantly at the tumor periphery, correlating with poor prognosis [75]. In these same peripheral zones, aggressive mesenchymal-like tumor cells interact with specific CAF subpopulations, particularly myofibroblastic CAFs, influencing immunotherapy response and patient survival [76]. The abundance of Tregs (FOXP3+) and CAFs expressing markers such as transgelin and fibroblast activation protein (FAP) in CCRCC has been associated with the presence of exhausted cytotoxic T lymphocytes (CD8+/PD-1+) in the tumor microenvironment, poor prognosis, and unfavorable responses to immune checkpoint inhibitors (ICIs) [76,83,84].

At this point, unveiling the precise profile of ITH seems mandatory in clinical settings [85]. Total tumor sampling is the gold standard procedure, but many tumors are too large, some RCC, for instance, so a total sampling of these cases is not a realistic option in routine practice. A multisite tumor sampling methodology based on the divide-and-conquer algorithm [86] has been successfully developed in routine practice to overcome this hurdle. This methodology couples clinical affordability and diagnostic efficiency in CCRCC [87] with applicability to other large neoplasms [88].

## 10. Intratumor Microbiome and Its Influence in Urologic Tumor Aggressiveness

The symbiotic dynamic interactions between bacteria and other microorganisms with human cells have been known for decades [89]. In this vein, many studies analyzing their influence on cancer evolution have demonstrated the presence of complex interconnections between the microbiome, cancer cells, and tumor microenvironment [90]. In this chapter, we focus strictly on urological cancer.

Some studies using 16S DNA sequencing have shown that RCC and normal renal tissue differ significantly in their bacterial composition [91]. Importantly, the microbiome alters important biological functions in tumor cells, such as membrane transport, transcription, and cell growth, while in non-tumor cells, among other functions, it modifies cell motility, signal transduction, and energy metabolism. The influence of the microbiome on the expression of PD-L1 in normal tissue, tumor, and tumor thrombus has also been analyzed [92].

UC has been a test bench for analyzing the effect of the local microbiome on tumor genesis and development [93]. Several studies have found that UC displays a decreased microbial diversity compared with non-tumor tissue and tobacco smoking seems to be involved in this change [94]. A conclusion could be that specific microbial signatures might be used to detect and/or monitor UC. On the other hand, a classically well-known precursor of bladder squamous cell carcinoma is Schistosoma *sp.* infection through inflammatory/immunological reactions.

Changes in gut bacteria [95] and the circulating fungal [96] microbiome have been extensively investigated in PCa patients. As specific changes in the urine microbiome are associated with high-grade PCa and PCa biochemical recurrence [97], urine analyses could potentially serve as tumor biomarkers, with diagnostic and prognostic implications.

Seminal plasma analysis is an accessible, non-invasive method by which to obtain specimens to investigate the microbiome of TC. However, very few studies have been performed on germ cell tumors of the testis so far [98], so the topic is open for promising future studies.

## 11. Ecological Principles and Mathematics Applied to Urological Cancer Study

Considering neoplasia as a huge community of individuals (cells) permanently interacting with each other, either cooperating or competing, it represents one of the most important steps in the study of cancer in recent times. From this ecological perspective, many tumor behaviors can be better understood and predicted, allowing for the development of new therapeutic approaches. The number of studies on different cancers analyzed using this viewpoint is rapidly growing, and the term “eco-oncology” has been coined [99]. In this context, interactions among tumor cells themselves are only a part of the general picture, since the role of the microenvironment and tumor/non-tumor cell interactions seems also crucial. Tumor-infiltrating lymphocytes [100], tumor-associated macrophages [101], and tumor-associated fibroblasts [102] have been demonstrated to play a preeminent role in tumor evolution and treatment.

From this perspective, up to four different tumor evolution types have been identified [103], i.e., linear, neutral, branching, and punctuated. In the linear evolution, driver mutations are so selective in that they outcompete all previous clones, giving rise to an intratumor quasi-homogeneity following a Darwinian model of evolution. Instead, in neutral evolution, new clones coexist without inter-clonal competition, giving rise to extreme levels of ITH following a non-Darwinian model. Branching tumors are models where clones developed from a common ancestry evolve in parallel and expand simultaneously, giving rise to a tumor with high levels of ITH following a Darwinian model of evolution. Finally, the punctuated type of evolution is characterized by the development of a very aggressive clone in the early phases of tumor development, generating a tumor with low levels of ITH, also following a Darwinian model. The interesting point of these conceptual types of tumors is that they correlate quite well with tumor progression and prognosis in clinical practice [104].

Most CCRCCs display either branching or punctuated evolutions, as was reflected in a multiregional genomic analysis with specific clinical behavior [8]. Branching-type tumors showed high ITH and an attenuated clinical course, with late and single metastasis, while punctuated-type ones behaved aggressively, with multiple and early metastases, and low levels of ITH. These molecular data correlate well with histological findings [105].

To note, ITH is not randomly distributed throughout the tumor. The tumor interior areas, where hypoxia and nutrient scarcity are high, develop aggressive clones with metastatic competencies, whereas the tumor periphery, where oxygen and nutrient supply is at its maximum, develops invasive capabilities at a local level [10,106]. As ecology rules predict, local/regional environmental pressures influence changes in the cellular genomics of cells in a fight for survival [107].

Coupled with ecology, applied mathematics has significantly improved our knowledge on cancer intricacies [108]. The use of game theory, for example, has explained why tumor cells behave as they do, thus demonstrating the hidden reasons for ITH development, thus predicting future behaviors that may eventually guide therapy [109,110]. For example, recent data suggest that histological and genomic data converge with game theory modeling using the hawk-dove game in analyzing the implications and significance of ITH in CCRCC [111].

Mathematics and ecology have also provided some plausible explanations and possible solutions to the problem of tumor resistance to therapy. As an example, some authors have proposed that a strategy of tumor containment using drugs under the maximum tolerable dose could benefit patients [112]. The ecological explanation of this anti-intuitive approach sustains that tumor cells developing under normal conditions spend all their energy on growing; therefore, a treatment following the maximum tolerable dose regime will force tumor cells to devote all their energy expenditure to generating resistances in order to survive. Sooner or later, a subset of tumor cells will attempt to develop resistances, and this fact will give rise to a uniformly resistant tumor. Since the amount of energy available in the cell is limited, a dose under the maximum tolerable threshold will make tumor cells divide their energy expenditure into two mutually exclusive tasks: to grow and to cope with the drug. The final result in this scenario is that both tasks, tumor growth and the development of resistance to therapy, will necessarily slow down, thus allowing retardation in both of them. This argument is the basis of understanding the Parrondo’s paradox applied to cancer [113], which reads that the combination of two losing strategies working together may win.

## 12. Conclusions

Urological cancer is a health problem in Western countries and represents an excellent test bench for basic and clinical research, as reflected in the literature. The spectrum of neoplasms in this field is broad and shows different pathogenic mechanisms with morphological, clinical, and therapeutic repercussions. This narrative intends to identify and review the hottest topics in the field, updating them for urologists, oncologists, and pathologists. Future perspectives will include further studies on some of the critical issues mentioned here, particularly the application of artificial intelligence, mathematics, and ecology to cancer; better knowledge of intratumor heterogeneity and its influence on treatment success; and the influence of tumor microbiomes on tumors.

## Figures and Tables

**Figure 1 cancers-17-01173-f001:**
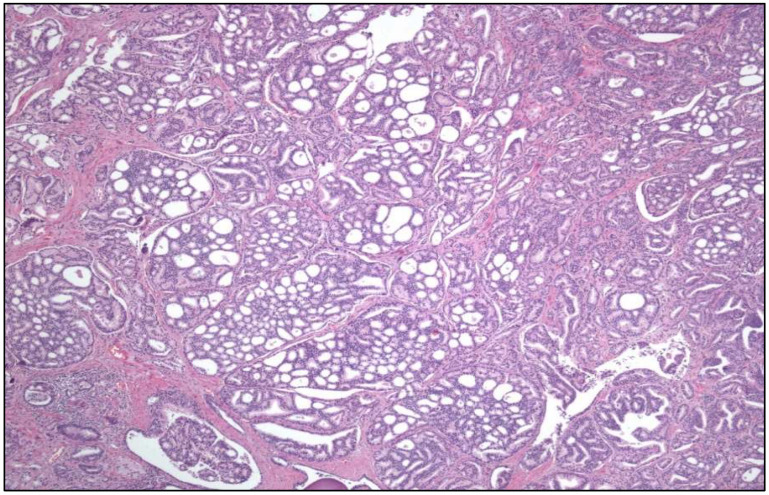
Big cribriform structures indicative of poorer prognosis in a prostate adenocarcinoma (in these cases, the lack of basal cells must be confirmed by immunohistochemistry).

**Figure 2 cancers-17-01173-f002:**
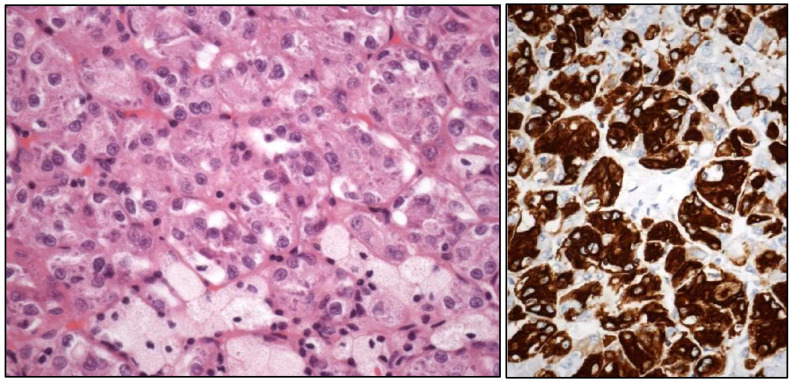
Hematoxylin-eosin staining of a eosinophilic solid and cystic renal cell carcinoma with its characteristic CK20 positivity (**right**).

**Figure 3 cancers-17-01173-f003:**
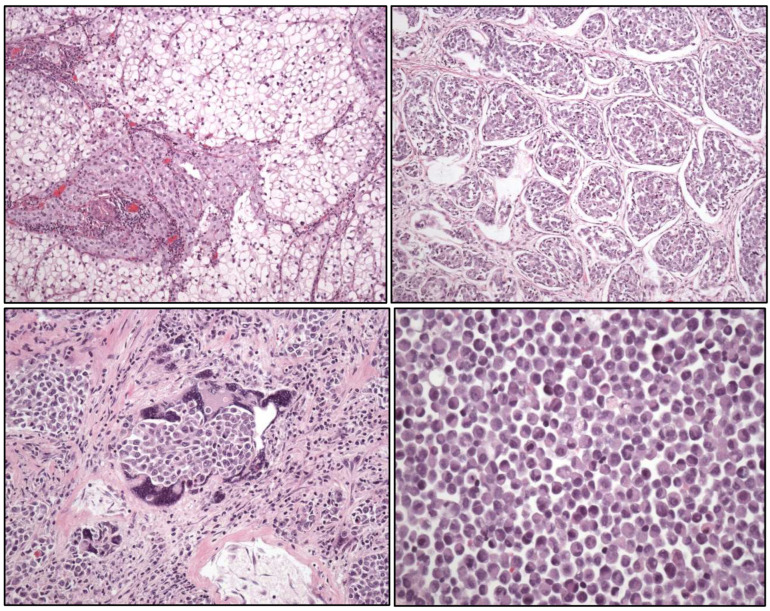
Histologic images of a clear cell (**upper left**), large nested (**upper right**), trophoblastic differentiation (**lower left**), and plasmacytoid (**lower right**) urothelial carcinoma of the urinary bladder.

**Figure 4 cancers-17-01173-f004:**
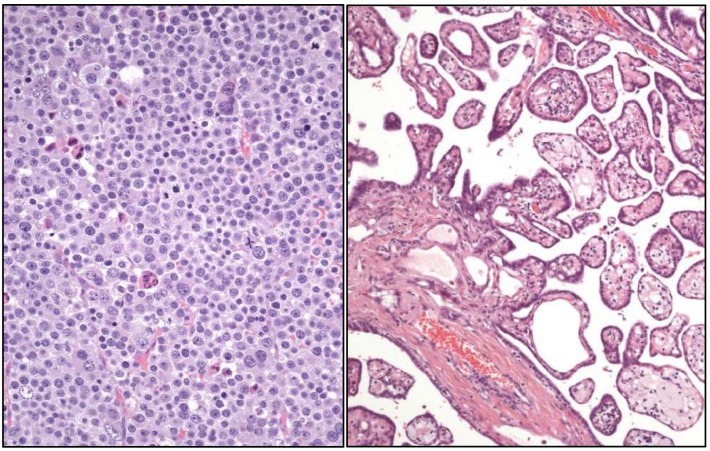
Typical histology of spermatocytic tumor (**left**) (original magnification, ×250) and well-differentiated papillary mesothelial tumor (**right**).

**Figure 5 cancers-17-01173-f005:**
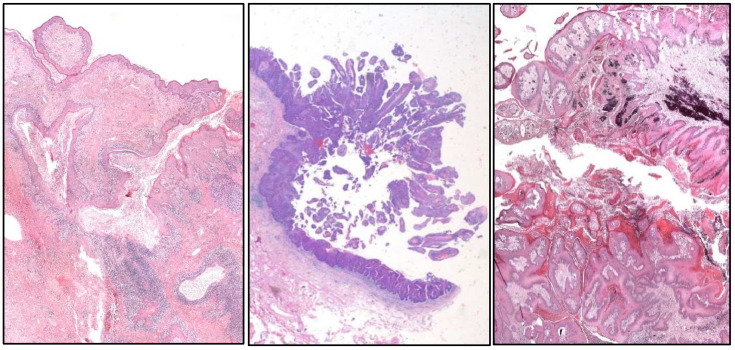
Histological features of “cuniculatum” (**left**), papillary (**center**), and verrucous (**right**) squamous cell carcinomas.

**Figure 6 cancers-17-01173-f006:**
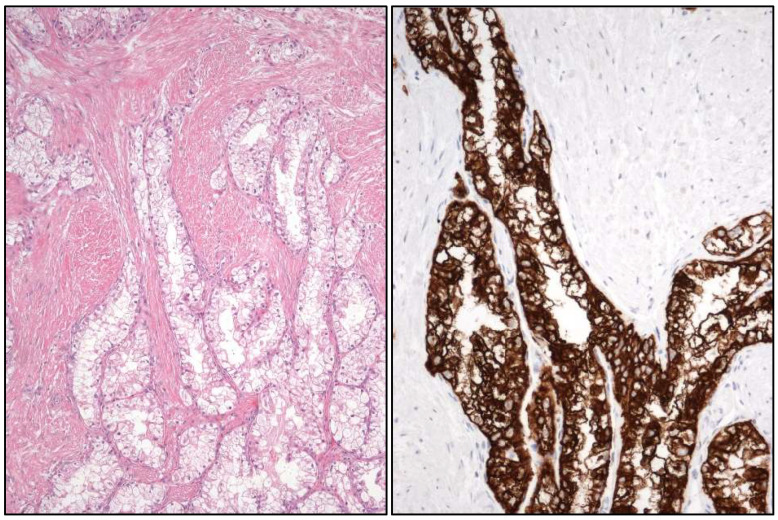
Histological (hematoxylin-eosin) and immunohistochemical (CK7+) (**right**) views of RCC with fibromyomatous stroma.

**Figure 7 cancers-17-01173-f007:**
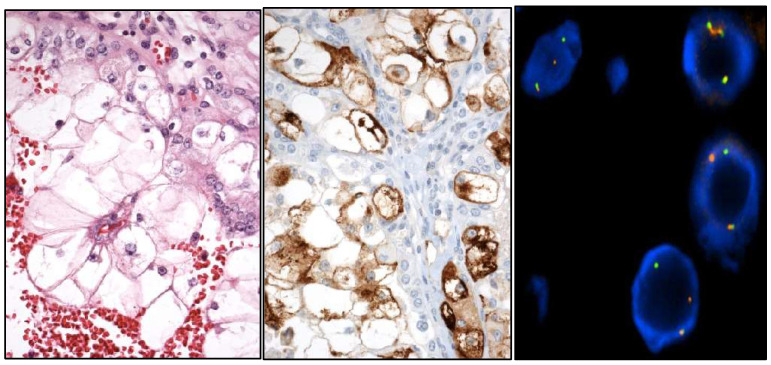
Histological view of a TFEB [t(6;11)] translocated renal cell carcinoma (**left**) with HMB-45 positivity (**center**) and FISH image (**right**).

**Figure 8 cancers-17-01173-f008:**
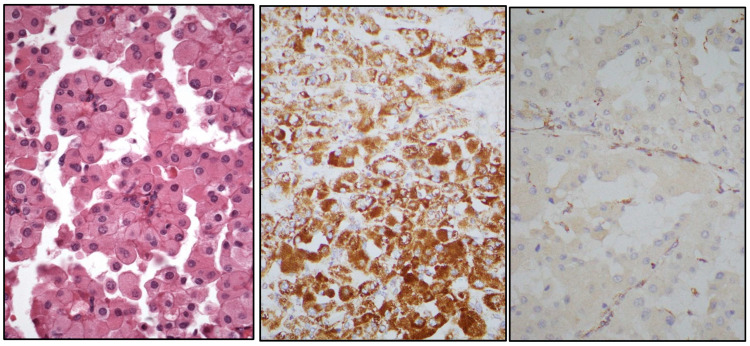
Histological view of a SDHB-deficient renal cell carcinoma (**left**) with retained SDHA (**center**) and lost SDHB (**right**) proteins.

**Figure 9 cancers-17-01173-f009:**
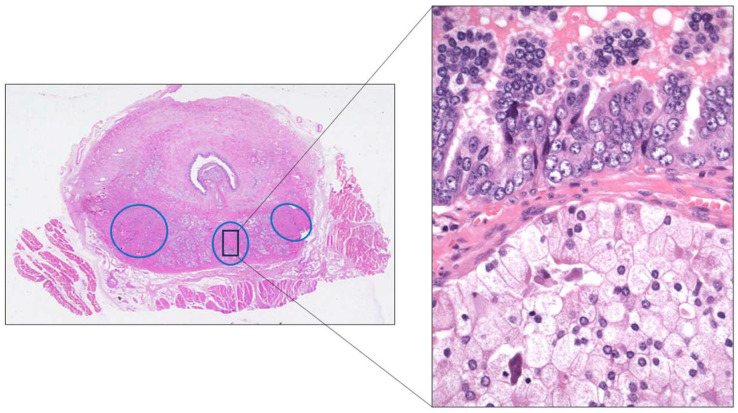
Example of prostate adenocarcinoma multifocality (blue circles) and intratumor heterogeneity (black rectangle) in an autopsy case.

**Figure 10 cancers-17-01173-f010:**
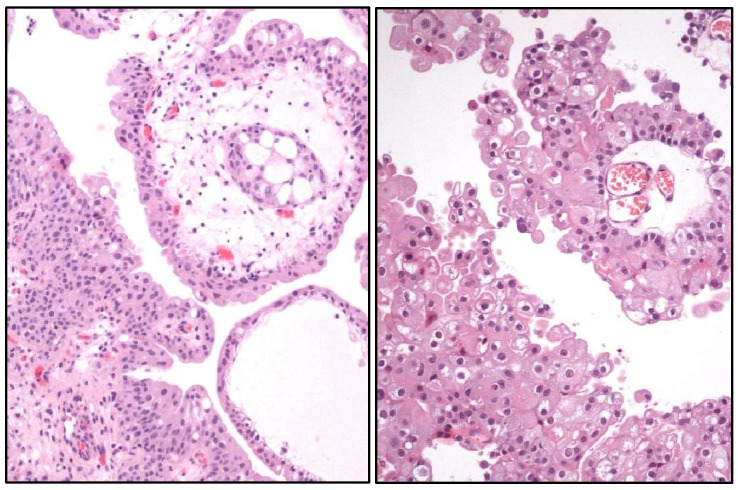
Histological image of a urothelial papilloma (**left**) and a papillary urothelial carcinoma (**right**).

**Figure 11 cancers-17-01173-f011:**
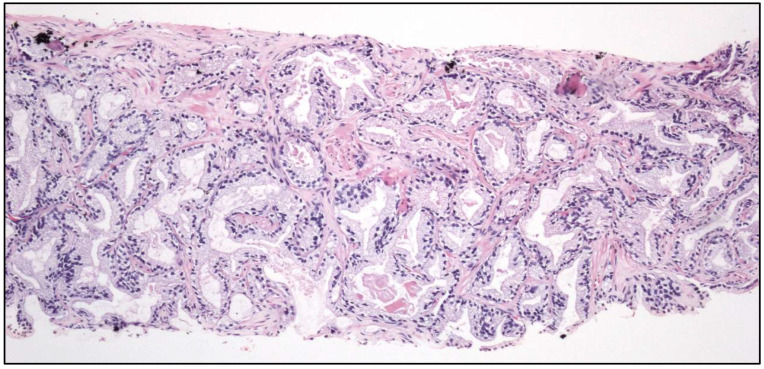
Gleason score 3 + 3 = 6 (Group 1) prostatic adenocarcinoma in a core biopsy sample.

**Figure 12 cancers-17-01173-f012:**
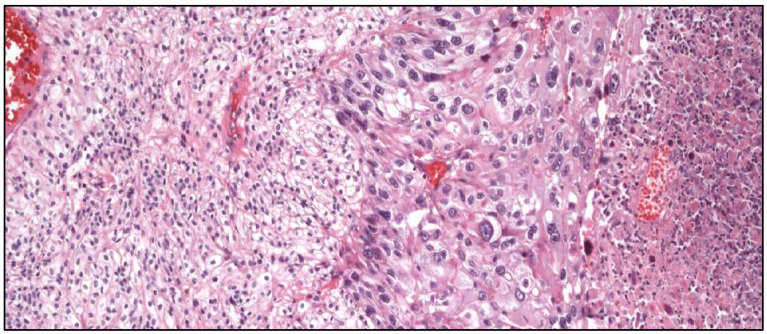
Histologic example of intratumor heterogeneity in clear cell renal cell carcinoma showing low (**left**) and high (**right**) grades.

**Table 1 cancers-17-01173-t001:** Hot spots in urogenital cancer research and clinics.

2022 WHO update on the classification of urinary and male genital tumors
New entities in kidney cancer
Urinary cancer-omics
Gleason grading system (update)
Targeted therapies and other novel treatments in urologic cancer
News on non-muscle invasive urothelial carcinoma
Artificial intelligence in urologic cancer
Intratumor heterogeneity’s influence on therapeutic failures in urologic neoplasms
Intratumor microbiome and its influence on urologic tumor aggressiveness
Ecological principles and mathematics applied to urological cancer study

**Table 2 cancers-17-01173-t002:** Molecular, histological, immunohistochemical, and gene expression in urothelial carcinoma.

**Luminal** (positive KRT20, GATA3, and FOXA1 markers) Luminal-papillary Papillary Histology (*FGFR3* mutation/fusion/amplification) Luminal-infiltrated positive EMT markers (TWIST1, ZEB1, etc.) Myofibroblast markers Wild-type *p53*
**Basal-squamous** (positive KRT5, KRT6, and KRT14 and negative GATA3 and FOXA1) Carcinoma in situ Squamous differentiation Basal keratin markers (CK 5.6, etc.) Immune inflammatory infiltrates (PD-1, PD-L1, etc)

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
