# Peer review of "Hot Spots in Urogenital Basic Cancer Research and Clinics"

_cancers, 2025, doi:10.3390/cancers17071173_

Round 1

Reviewer 1 Report

Comments and Suggestions for Authors

A very extensive study on the important problem of urological cancers. However, the knowledge contained in the article is widely known and generally easily accessible. I believe that Chapter 2. Methods is unnecessary. The aim of the work is presented earlier, and the journals and articles on which the authors rely are given in References (a significant part of the literature, if I counted correctly - 20 items are self-citations).

Author Response

According to the reviewer’s comments, the Chapter 2 has been removed and the Table 1 has been referenced in the Introduction. Self-citations in a review paper where the authors have worked for years amounting a nice experience on the topic should not be a problem. This is not an original paper and no new information is given.

Reviewer 2 Report

Comments and Suggestions for Authors

This is a well-written article. It covers hot spots in urological basic cancer research and clinics with many figures on tissue biopsy analyses. However, it is strange that the subject of liquid biopsy has not been found as one of hot spots. It would be more interesting if authors can intergrate the advancements of liquid biopsy to complement with the tissue analysis in the field of urological cancer research and clinics.

Author Response

We understand the point. However, the literature review performed to identify the current hot topics in urogenital oncology did not point to the liquid biopsy and some other interesting topics as a “hot topic”. We had to write a manuscript with limits which we defined aprioristically, and liquid biopsy applied to urogenital oncology was not there.

Reviewer 3 Report

Comments and Suggestions for Authors

1) General comments

In this review article, recent topics of urogenital cancers were summarized. The authors well described this issue, but the reviewer has some comments indicated below.

2) Specific comments

  1. Although title of the present review only includes “urological”, there are descriptions regarding prostate and testicular cancers. “Urogenital cancer” would be better wording.
  2. Please use “Gleason score” instead of “Gleason index”.
  3. Line 96: “adenocarcinoma with ductal features” instead of “,,with ductal changes”.
  4. Lines 103-104: PSA and prostatic acid phosphatase should be negative in treatment-related prostatic neuroendocrine carcinoma.
  5. Figure 1: the present lesion seems to include significant component of intraductal carcinoma (IDC). Loss of basal cells should be confirmed by immunohistochemistry.
  6. Lines136-137: Differentiation of pTa (non-invasive) disease from pT1 (invasive) disease in low-grade papillary urothelial carcinoma is mandatory. The reviewer could not understand this sentence.
  7. Lines 167-169: This is not only a rule in testicular cancer, but a general rule in all kinds of cancers (WHO 5th edition).
  8. “Conclusions” sound like just introduction. Please simply describe the significance of this review and future prospectives in this research field.

Author Response

  1. Although title of the present review only includes “urological”, there are descriptions regarding prostate and testicular cancers. “Urogenital cancer” would be better wording.

Yes, “Urological cancer” has been changed by “urogenital cancer”.

  1. Please use “Gleason score” instead of “Gleason index”.

“Gleason index” and “Gleason score” are synonyms widely used in prostate cancer pathology. Following the reviewer suggestion, we have changed it using “Gleason score” in the text. 

  1. “adenocarcinoma with ductal features” instead of “,,with ductal changes”.

Yes, “ductal features” terminology has been introduced instead of “ductal changes”

  1. PSA and prostatic acid phosphatase should be negative in treatment-related prostatic neuroendocrine carcinoma.

Yes, the reviewer is totally right and the sentence “Noteworthy, PSA and prostatic acid phosphatase are usually negative in treatment-related neuroendocrine carcinoma” has been added in the text.

  1. Figure 1: the present lesion seems to include significant component of intraductal carcinoma (IDC). Loss of basal cells should be confirmed by immunohistochemistry.

Yes, we have completed the Figure 1 legend including “(in these cases the lack of basal cells must be confirmed by immunohistochemistry).

  1. Lines136-137: Differentiation of pTa (non-invasive) disease from pT1 (invasive) disease in low-grade papillary urothelial carcinoma is mandatory. The reviewer could not understand this sentence.

The 2022 WHO edition recognizes the frequent impossibility of distinguishing pTa from pT1 in non-muscle-invasive urothelial carcinomas in transurethral resection specimens. Specifically in these cases, an impossible task remains impossible to perform, and for this reason it is not mandatory (although advisable). This frequent impossibility is due to several factor, for example the specimen misorientation and the frequent artifacts detected in transurethral resection specimens. To clarify this point we have included in the text the following specification “It is not mandatory (although advisable when possible) to define the level of invasion (pTa vs. pT1) in the low-grade non-invasive papillary urothelial carcinoma (UC)”.  

  1. Lines 167-169: This is not only a rule in testicular cancer, but a general rule in all kinds of cancers (WHO 5th edition).

Yes, the reviewer is right.

  1. “Conclusions” sound like just introduction. Please simply describe the significance of this review and future prospectives in this research field.

Yes, done.